# Effects of Zn, Mg, and Cu Content on the Properties and Microstructure of Extrusion-Welded Al–Zn–Mg–Cu Alloys

**DOI:** 10.3390/ma16196429

**Published:** 2023-09-27

**Authors:** Krzysztof Remsak, Sonia Boczkal, Kamila Limanówka, Bartłomiej Płonka, Konrad Żyłka, Mateusz Węgrzyn, Dariusz Leśniak

**Affiliations:** 1Lukasiewicz Research Network—Institute of Non-Ferrous Metals, 44-100 Gliwice, Poland; 2Faculty of Non-Ferrous Metals, AGH University of Krakow, 30-059 Krakow, Poland

**Keywords:** extrusion welding, Al–Mg–Zn–Cu, Al7075, alloying elements, EBSD, GOS

## Abstract

The study presents the results of research on the influence of different contents of main alloying additions, such as Mg (2 ÷ 2.5 wt.%), Cu (1.2 ÷ 1.9 wt.%), and Zn (5.5 ÷ 8 wt.%), on the strength properties and plasticity of selected Al–Zn–Mg–Cu alloys extruded on a bridge die. The test material variants were based on the EN AW-7075 alloy. The research specimens, in the form of 100 mm extrusion billets obtained with the DC casting method, were homogenized and extrusion welded during direct extrusion on a 5 MN horizontal press. A 60 × 6 mm die cross-section was used, with one bridge arranged in a way to extrude a flat bar with a weld along its entire length. The obtained materials in the F and T6 tempers were characterized in terms of their strength properties, hardness, and microstructure, using EBSD and SEM. The extrusion welding process did not significantly affect the properties of the tested materials; the measured differences in the yield strength and tensile strength between the materials, with and without the welding seam, were up to ±5%, regardless of chemical composition. A decrease in plasticity was observed with an increase in the content of the alloying elements. The highest strength properties in the T6 temper were achieved for the alloy with the highest content of alloying elements (10.47 wt.%), both welded and solid. Significant differences in the microstructure between the welded and solid material in the T6 temper were observed.

## 1. Introduction

Precipitation-hardened Al–Zn–Mg(Cu) alloys are characterized by their high strength, ductility, elastic modulus, corrosion resistance, as well as their fracture toughness. For this reason, they are widely used in the aerospace industry (airframe, fuselage), automotive industry (bumpers, body parts), among manufacturers of sports equipment, and in other industries where weight reduction while maintaining high-strength properties is important [1,2,3]. However, the 7XXX series aluminium alloys are also highly sensitive to the strain rate and deformation temperature, which directly translate into low processing efficiency, especially during the extrusion of welded profiles (on bridge or porthole dies) [4]. This is due to phenomena that produce a hot cracking effect at elevated working rates [5]. During the extrusion process, both in the press container and in the die itself, frictional forces arise that disturb the favorable state of hydrostatic stresses [6]. These forces cause tensile stresses to accumulate on the surface of the extruded band and cause additional heating of these areas. In extreme cases, local temperature increases in the material and the resulting stress release may result in the loss of continuity of the extruded strand [7]. This is of particular importance in the case of bridge–chamber dies, where the complicated internal geometry multiplies the adverse effects of frictional forces on the homogeneity of the material outflow [8]. The structural phenomena accompanying the extrusion process on porthole dies should also be considered. In the studies carried out so far, it was found that in the 6063 aluminium alloys extruded on porthole dies, the occurrence of dynamically recrystallized (DRX) grains was observed both in the welded area and beyond it. It was also noted that the amount of the recrystallized fraction may be related to the differentiation of the strain rate on the cross-section of the extruded profile [7]. Similar conclusions apply to Al–Zn–Mg alloys extruded on porthole dies. It has been shown that a higher strain temperature and a lower strain rate have a positive effect on the occurrence of dynamic recrystallization. Moreover, in the case of Al–Zn–Mg alloys, it was proven that in both the areas close to the walls of the profiles and in the welding zones, the recrystallized fraction is higher than in other zones [9]. Studies conducted on extruded Mg–Al–Zn alloys also indicate that complete dynamic recrystallization occurred in the welding areas [10]. Current studies on Al–Zn–Mg alloys extruded on porthole dies show a significant differentiation in the microstructure between the welded area and the outside of the weld [11].

The material for the extrusion of aluminium alloys are billets, cast using the DC (di-rect chill) method. It is a method of semicontinuous casting into a water-cooled crystal-lizer. The distinctive feature of that process is directional crystallization, taking place from the outer zones (intensively cooled zones near the crystallizer) to the center of the ingot. As a result, a dendritic structure is formed [12]. Dendrites from the Al-rich α phase grow along the temperature gradient. This directionality of solidification leads to microsegregation, and coarse intermetallic phases are formed that can significantly affect the properties and susceptibility to hot working, which directly affects the efficiency of technological processes. The above-mentioned segregation in Al–Zn–Mg(Cu) alloys during casting results in high concentrations of Cu, Mg, and Zn in interdendritic eutectic regions. This has a significant impact on reducing the corrosion resistance of the alloys, and can also be a place of crack initiation of the billets themselves [13]. In the case of Al–Zn–Mg(Cu) alloys, literature sources indicate that the main phases present in the material after casting are η-MgZn2, T-Al2Mg3Zn3, S-Al2CuMg, and θ-Al2Cu. The S, T, and η phases described in the literature are solid solutions with extended composition ranges, containing all four elements. It should be noted that the literature data clearly indicate that the mentioned phases are low-melting, especially η-MgZn2, and their presence causes a significant decrease in the solidus temperature, even below 480 °C for the 7075 alloy [12,13]. It causes a large difference between the liquidus and solidus temperatures, which in turn leads to the need to lower the temperature of plastic-forming processes. As a result, it translates into a decrease in the speed of these processes (e.g., extrusion) and a decrease in the efficiency. Homogenization annealing is a way to increase the solidus temperature of Al–Zn–Mg(Cu) alloys. Data from the literature clearly indicate that the dendritic structure gradually dissolves during homogenization. Diffusion into the matrix in eutectic structures and gradual dissolution of the Mg (Zn,Cu,Al)2 phase takes place. From the literature data, the conclusion drawn was that the driving force of phase transformations from Mg (Zn,Cu,Al)2 to Al2CuMg must be the supersaturation of copper in areas of the eutectic structures, which makes Al2CuMg a stable phase [12,13].

In the case of precipitation-hardened 7XXX series alloys, the content, distribution, and size of the matrix precipitates have a key impact on their strength. It can be increased by increasing the content of the elements dissolved in the matrix during solution treatment, which can be separated from the solution in the aging stage [14,15]. During the supersaturation process, it is necessary to ensure sufficiently rapid cooling, which will ensure a uniform distribution of precipitates of the fine-grained n’ phase during aging. Since the metastable n’ phase should have the greatest impact on precipitation strengthening, the greatest strengthening effect will be obtained as a result of rapid cooling [16,17].

Thus far, the influence of the main alloy additions, that is, Zn, Mg, and Cu, on the strength properties of the 7XXX series alloys has been described in the literature [15,18,19,20,21]. For alloys with a constant Cu and (Zn+Mg) content, an increase in the Zn content will increase the strength of aged samples [15,18]. It was also found that with a constant Cu content, an increase in the share of (Zn+Mg) in the T6 temper will result in an increase in the number of precipitates, which will translate into an increase in the strength of Al–Zn–Mg–Cu alloys [17]. Of equal importance is that this can result in a large difference in plasticity between the matrix and PFZ areas (precipitation free zones), and consequently may lead to a reduction in fracture toughness also caused by coarse slip [17].

Al–Zn–Mg–Cu alloys, due to technological difficulties, are rarely used in extrusion processes on porthole dies. At the same time, the industry is systematically increasing its interest in high-strength Al alloys, also in the form of thin-walled closed profiles. For this reason, as well as due to the lack of relevant publications, there was a need to conduct research on the effects of different levels of the main alloying additives on extrusion-welded Al–Zn–Mg–Cu alloys. 

This study investigated the influence of different contents of main alloying additions on the strength properties and plasticity of selected Al–Zn–Mg–Cu alloys extruded on bridge dies.

## 2. Materials and Methods

The tests were carried out on modified 7xxx series alloys, based on the 7075 alloy. 

The initial stage of the research included casting variants of EN AW-7075 alloys in the form of 100 mm diameter billets on a semicontinuous production line consisting of 300 kg Monometer resistance furnace and a direct chill casting crystallizer (Monometer House, Rectory Grove, Leigh-on-Sea, Essex SS9 2HN, UK).

The chemical composition of the alloys was analyzed using optical emission spectrometry, and is presented in Table 1. Table 2 shows the contents of the main alloying elements and their proportions.

During the casting process, the metal was filtered through a 30 ppi ceramic filter. The alloys were cast in stages, two billets at a time, each approx. 2 m long (Figure 1).

The billets were subjected to homogenizing annealing to maximize the solidus temperature level, which is of great importance in the context of maximizing the deformation rate of Al–Zn–Mg alloys [22,23]. This is important from the point of view of extrusion welding, as it significantly extends the permissible temperature range of the material in the deformation cavity. Homogenizing parameters were the subject of other research studies conducted [13]. The final annealing conditions are presented in Table 3.

In the next stage of the research, extrusion of the flat bar on a die with a single bridge was carried out. The extrusion was performed on a 5 MN (500 t) horizontal hydraulic press (ZAMET BUDOWA MASZYN S.A., 83 Zagórska Str., 42-680 Tarnowskie Góry, Poland) in direct mode with extrusion force registration. For this purpose, a 60 × 6 mm die insert was designed and made for the existing tooling set, with a single bridge placed perpendicularly to the long edge of the extruded flat bar to allow the formation of a longitudinal weld in the middle of its width (Figure 2). The 60 × 6 mm flat bar was extruded from all variants of the 7075 alloy. The material was prepared in the form of Ø100 × 200 mm billets, then heated to 500 °C for extrusion, with a ram velocity of 1 mm/s. Table 4 presents the registered peak forces during the extrusion process.

The extruded sections were subjected to heat treatment of the T6 temper for all of the alloy variants. For this purpose, samples for solution treatment and artificial aging were taken from the extruded bars. The heat treatment parameters are listed in Table 5. To verify the effects of the heat treatment, a Brinell hardness test was performed. Based on the results obtained, the aging curves presented in Figure 3 were made for the individual variants of the alloys. They show that the tested variants of the alloys obtained more than 90% of their maximum hardness values after about 8 h. Further aging up to 24 h increases the hardness, but for industrial use it may be more economical to shorten the aging time. Therefore, for the purposes of this study, the aging time was set at 8 h.

The heat-treated material and the reference material in the F temper were intended for further research.

A static tensile test was performed in accordance with the requirements of the standard PN-EN ISO 6892-1:2020-05 [24] on the Instron 5582—max load 100 kN. The strain deformation was measured with an extremely accurate video extensometer (Instron, Norwood, MA, USA), with crosshead speed 1 = 0.37 mm/min and crosshead speed 2 = 3 mm/min. The gauge length was 20 mm. A Brinell hardness test was conducted according to PN-EN-ISO-6506-1_2014 [25] on Duramin 2500E hardness tester (Struers, Ballerup, Hovedstaden, Denmark), with a ball diameter 2.5 mm, main load 31.25 kgF, and expanded uncertainty with the confidence of the result at the 95% level with the coefficient k = 2, determined indirectly by the M2 method in accordance with the PN-EN-ISO-6506-1_2014 standard.

The samples for the static tensile test were taken across the flat bar in such a way that the welded area was located in the middle of the length of the measurement base. The reference material was taken from a flat bar that was extruded without a weld. The scheme of taking strength samples is presented below (Figure 4). The hardness test was carried out on the cross-section of the flat bars, in the middle of their thickness, with measurement points (1–13) placed every 5 mm, and in the central zone every 2.5 mm, as shown in Figure 5.

The microstructural characterization of the alloys was carried out with a high-resolution INSPECT F50 FEI scanning electron microscope with attachments for the chemical analysis via EDS and a Velocity plus EBSD camera (FEI Company, Hillsboro, OR, USA). The EBSD analysis was performed using EDAX Apex Advanced (ver. 2.5.1001.0001) and EDAX OIM Analysis 8 software (ver. 8.6.0101x64) and ICCD 2011 PDF database format. The samples for the tests were ground on SiC abrasive papers up to 4000 grit, and polished with diamond suspensions up to 1 μm. The final operation was polishing with a colloidal SiO_2_ suspension with gradations of 0.01 μm to obtain a perfectly flat surface. The samples for crystallographic analysis were prepared on an RES101 ion milling instrument (Leica Microsystems, Wetzlar, Germany).

## 3. Results and Discussion

The extrusion load was recorded during the process, and the extrusion curves of the tested alloys were prepared (Figure 6). The highest yield resistance represented by an extrusion peak load up to 5.06 MN, was characteristic of alloy 4; it was slightly lower, 4.78 MN for alloy 2, then 4.40 MN for alloy 3 and 4.11 MN for alloy 1. It is clearly visible that the yield resistance depends on the level of the alloying components, which for alloy 4 is (Mg+Zn+Cu) 10.47%. In the case of alloys 2 and 3, for a similar level (Mg+Zn) of 8%, the decisive factor for the yield resistance is the Cu contents of 1.91% and 1.53%, respectively. The lowest yield resistance was recorded for alloy 1 with the lowest level of the main alloying elements (Mg+Zn+Cu), 8.12%.

In the next stage of research, hardness measurements were made on the cross-section of the extruded flat bars, both in the F and T6 tempers. The hardness distribution curves are shown below in Figure 7. The analysis of the results shows a clear increase in the hardness of all of the alloy variants after heat treatment of the T6 temper. The lowest increase was recorded for alloy 1 (60 HB), and the highest for alloy 4. A slight (up to 5 HB) hardness variation was observed in the welded area (half the width of the flat bars), indicating the uniformity of the strength properties of the flat bar.

In the case of the tested variants of the Al–Zn–Mg–Cu alloy, heat treatment of the T6 temper resulted in an increase in tensile strength TS (Figure 8) and yield strength YS (Figure 9), both for the welded areas and the solid material. The largest increase in the tensile and yield strength TS and YS was recorded for alloy 4, and amounted to 244 MPa and 353 MPa, respectively, for the unwelded samples, and 305 MPa and 352 MPa, respectively, for the welded samples. For the other alloys tested, the increases in strength properties were smaller, and amounted to 150–220 MPa for the TS and 250–300 MPa for the YS. Differences in the increases of the described properties are directly caused by the amount of alloy additions in the individual alloys tested. Alloy 4 contains the highest addition level of 10.47 wt.%, with the content of Zn alone being 7.76 wt.%. According to the literature, the content of this alloying additive contributes most significantly to the precipitation strengthening of Al–Zn–Mg–Cu alloys, as it directly translates into an increase in the amount of the MgZn2 phase, which is the main strengthening phase of these alloys. Heat treatment of the T6 temper did not significantly affect the differentiation of the strength properties in these areas of the individual alloy variants.

A clear decrease in elongation in the welded areas is noticeable for each of the tested alloy variants (Figure 10). The unfavorable effect of increasing the content of alloy additions on the elongation of the tested alloys can also be observed. Variants 2 and 3 have similar amounts of main alloying elements, which are 9.91 wt.% and 9.61 wt.%, respectively, and are characterized by a similar elongation level for both the welded and solid samples in the F and T6 tempers. The highest elongation in all of the variants tested was recorded for alloy 1, and the lowest for alloy 4. The exception was the sample in the T6 state, which had an elongation greater by more than 2% than variants 2 and 3, whose elongations in each case were similar.

Figure 11 shows SEM images of alloy 1 (a), alloy 2 (b), alloy 3 (c), and alloy 4 (d). The microstructure of alloys 3 and 4 have relatively smaller and more densely distributed fine precipitates in the Al matrix than those in alloys 1 and 2. All of the alloys reveal irregularly shaped and bright-coloured intermetallic particles. 

Figure 12 shows IPF images of the extruded alloys at the weld site from a cross-sectional view to the extrusion direction. The analyses were performed in the T6 temper. The EBSD analysis showed that the shape and size of the grain in the weld were different depending on the chemical composition of the alloys. In alloy 1 shown in Figure 12a, and in alloy 4 shown in Figure 12d, deformed and elongated grains with mainly the (001) and (111) orientations were analyzed. In the middle part of the microstructure, areas of equiaxed grains occurring within the boundaries of large, elongated grains were also observed. The grains occurred in a wide range of sizes, from 1 to over 200 mm. Meanwhile, the microstructures of alloys 2 (Figure 12b) and 3 (Figure 12c) were characterized by equiaxed grains with a random orientation, indicating the occurrence of dynamic recrystallization processes at the weld.

The distributions of the average grain size, shown as histograms in Figure 13, confirm the significant differences in structure for the four alloys studied. The largest grains were found in alloy 1, with the finest grains occurring in a range of up to 25 μm in alloys 2 and 3. Alloy 4 had a wider grain size distribution compared to alloys 2 and 3, ranging up to 45 μm.

To compare the structures in the alloys analyzed, analogical scans were taken in the areas outside the welds (Figure 14). It was found that the structures in the four alloys analyzed showed no significant differences. Alloy 1 showed a slightly larger grain size compared to the other alloys analyzed. The grain size distribution shown in Figure 15 for Alloy 1 shows a wider grain size range up to 55 μm. In alloys 2 to 4, the range was approximately 32 μm.

Figure 16 and Figure 17 represent the GOS maps obtained from the IPF images shown in Figure 12 and Figure 14. The GOS value represents the degree of average dispersion of the misorientation within the same grain in a welded area. Low GOS values can be obtained if some restoration of the microstructure, such as recovery or recrystallization, occurs in the chosen grain. Low GOS values (below 2° or 3°) have been proven to represent recrystallized areas [26]. As a result, as shown in Figure 16 and Figure 17, the blue grains have the smallest GOS values and indicate dislocation-free grains with orientations spread between 0 and 2. In the experiment, the interval with the smallest GOS values with a threshold of 2° represents the recrystallized grains, while the GOS value above 2° represents the deformed grains [27]. The red grains, with orientation spreads between 5 and 35, indicate the heavily deformed grains.

As a result of these analyses, it was found (Figure 16) that welded alloys 2 and 3 had the highest proportions of grains with recrystallized fractions. Alloy 1 was characterized by large partially deformed grains. Alloy 4, on the other hand, showed fine bar-shaped grains that were partially deformed. Recrystallized grains were observed in the very center of the weld. The GOS values for the areas outside the weld (Figure 17) for the four alloys tested were very similar. The highest proportion of grains with a recrystallized fraction was found in alloy 2. However, the differences in the orientation distributions between the alloys were not significantly large.

The low content of the main alloying elements in alloy 1 translates into the lower plastic resistance of this material. This is reflected in the level of the recorded force during extrusion of this alloy on the press (4.11 MN). The lower plastic resistance mentioned above results in lower friction forces in the die deformation cavity and, consequently, in a smaller increase in the temperature of the extrusion. This, in turn, results in the formation of a deformed structure in alloy 1 with a low share of recrystallized grains, which was about 15%. In this case, no effect of the Zr addition on the fragmentation of the structure was observed, which is also the result of an insufficient increase in temperature during the extrusion. In sample 1, dynamic recovery occurs, and the shape of the grain is deformed. 

Welded alloy 1 in the T6 temper had a tensile strength of about 540 MPa, and the elongation was up to 6%. The larger elongation may have been induced by the large grain size. In the case of alloy 1, the grain size distribution obtained from the EBSD test was disturbed. This is due to limitations of the scanned image analysis software, which does not take into account grains with unidentified boundaries. In this case, the largest surface fraction are for grains that are above 80 µm.

In the case of welded alloys 2 and 3, which nearly recrystallized, the T6 temper was also recorded with a tensile strength of about 540 MPa, but the elongation in that case was about 1%. This reduction in plasticity of alloys 2 and 3 relates to an inhibition of grain growth. The proportion of the recrystallized fraction in alloys 2 and 3 is approximately 81%. 

For alloy 4, a slightly higher elongation of about 3% was found in the weld compared to alloys 2 and 3. This could be the result of a higher internal stress level due to the highest content of alloying elements in this alloy. From the observations, the grains in alloy 4 were also elongated along the weld line. The grain size along this line was greater than 100 μm, and less than 10 μm in the transverse direction. During the static tensile test, which was conducted in a direction transverse to the weld line, the grains underwent greater plastic deformation than the equiaxial fine grains in alloys 2 and 3.

All of the alloys contain alloying additions of Mg, Cu, and Zn, and additionally Zr. These elements can form MgZn2 and Al3Zr phases, while Cu remains in solid solution. Numerous studies [28,29,30,31] confirm that additives such as Zr are inhibitors of recrystallization. The nano-sized dispersoids formed by Zr can pin down and block the migration of the grain boundaries and, as a result, inhibit the recrystallization in Al–Zn–Mg–Cu alloys [30]. The MgZn2 phase, on the other hand, has a strong precipitation strengthening effect for the Al–Zn–Mg–Cu alloys, and its degree is strongly dependent on the amount of Zn. The higher the Zn content in the alloy, the more the MgZn2 precipitates in the material after the aging treatment. In conclusion, by increasing the Zn content in the Al–Zn–Mg–Cu alloys, the strength is increased [28].

In the case of welded alloy 4 in the T6 temper, the recrystallized grains fraction on the GOS image was about 41%. The introduction of a large amount of the alloying elements resulted in the formation of fine and densely distributed dispersoids that had a significant influence on the recrystallization behavior in alloy 4. Some of the grains remained fibrous, while others recrystallized. This phenomenon is related to the orientation of the grains, where the recrystallized grains are <111> oriented and the fibrous grains are <100> oriented; this was also confirmed by literature data [28]. The tensile strength of alloy 4 was almost 650 MPa, while the elongation was about 4%. The characteristic grain structure arrangement, perpendicular to the extrusion direction, was a result of the specific conditions in the welding chamber of the extrusion die. Consequently, there was a directional flow of the metal and vertical arrangement of the grains in the areas close to the weld.

## 4. Conclusions

Heat treatment of the T6 state of the tested materials resulted in a significant increase in the hardness of all of the alloy variants. The lowest increase was recorded for alloy variant 1 (60 HB), and the highest for alloy 4.There was no significant differentiation in hardness in the area of the weld (half the width of the flat bars), which indicates the uniformity of the strength properties of the flat bar. Alloys containing Cu, both in the F and T6 states, are characterized by similar strength properties. The presence of a weld does not significantly affect their properties.The highest strength properties (TS, YS, HB) characterize samples of alloy 4 in the T6 temper. The highest elongation (A) was found in alloy 1 (low level of alloying elements) in each tested variant.In the T6 temper, in alloys with a similar Cu content (alloys 3 and 4), an increase in the Zn content by 2% resulted in increases in the strength, yield strength, and elongation. For alloys with a similar Zn content (alloys 2 and 3), the increase in properties (TS, YS, A) is determined by the Cu content.A clear decrease in elongation in the samples examined across the weld was noted for each material tested.It was found in the T6 temper that the microstructure of extrusions without the weld in the four alloys analyzed showed no significant differences.Analysis of the microstructure of welded extrusions in the T6 temper confirmed the significant differences in the grain size distribution. The largest grains were found in alloy 1, with the finest grains occurring in a range of up to 25 μm in alloys 2 and 3. Alloy 4 had a wider grain size distribution compared to alloys 2 and 3, ranging up to 45 μm.A high solute content of the main alloying elements in Al–Zn–Mg–Cu alloys, as well as the presence of Zr, can lead to a high resistance to recrystallization. A large amount of the alloying elements resulted in the formation of fine and densely distributed dispersoids that had a significant influence on the recrystallization behavior in alloy 4. Some of the grains remained fibrous while others became recrystallized. This phenomenon is related to the orientation of the grains, where the recrystallized grains are <111> oriented, and the fibrous grains are <100> oriented.

## Figures and Tables

**Figure 1 materials-16-06429-f001:**
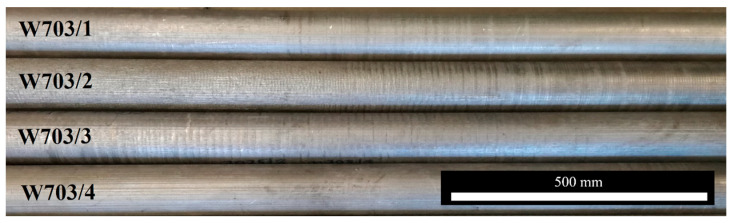
Cast billets of the investigated alloys. From the top to the bottom: alloy 1, 2, 3, 4. W703 is the casting batch number.

**Figure 2 materials-16-06429-f002:**
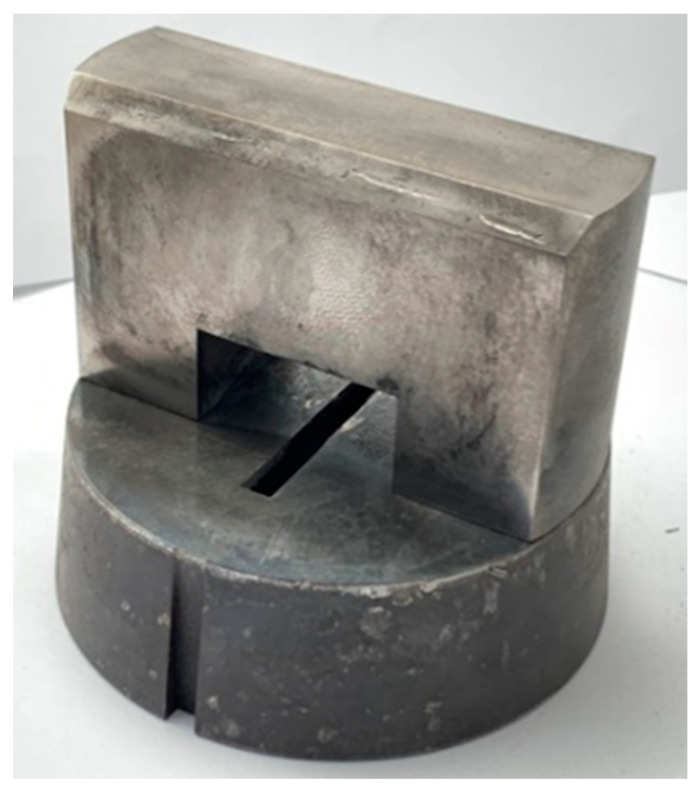
A die insert with single bridge used for extrusion of 60 × 6 mm flat bar with weld.

**Figure 3 materials-16-06429-f003:**
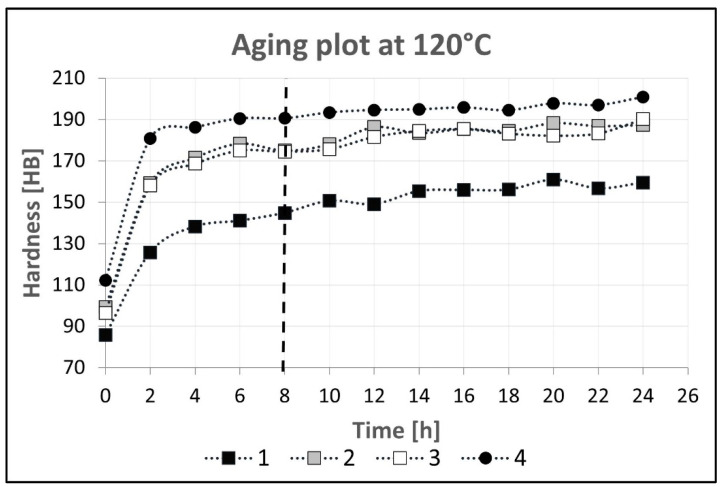
Aging plot for investigated alloys.

**Figure 4 materials-16-06429-f004:**
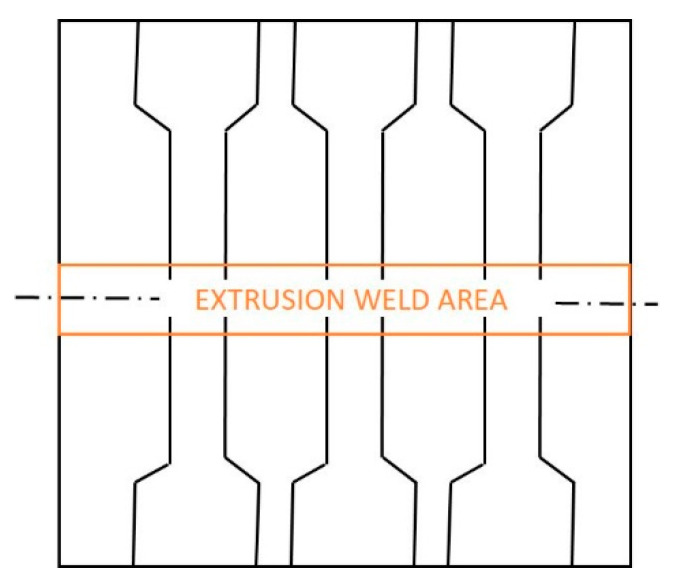
An example of sampling for static tensile test.

**Figure 5 materials-16-06429-f005:**
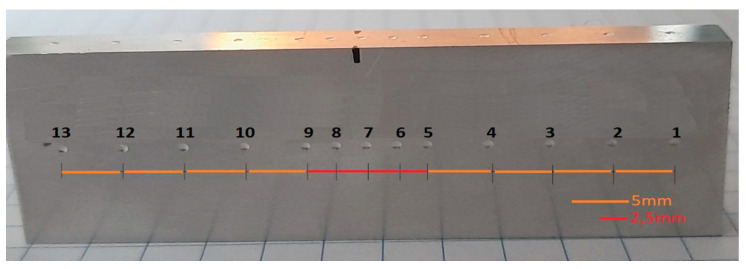
An example of the distribution of Brinell hardness test measurement points.

**Figure 6 materials-16-06429-f006:**
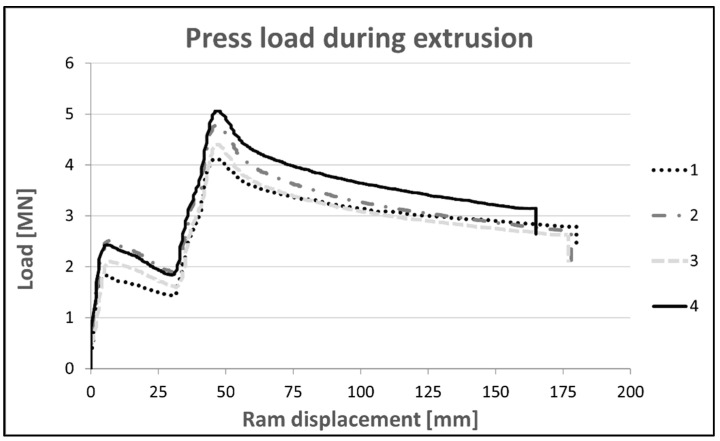
Press loads recorded during 60 × 6 mm section extrusion of tested aluminum alloys.

**Figure 7 materials-16-06429-f007:**
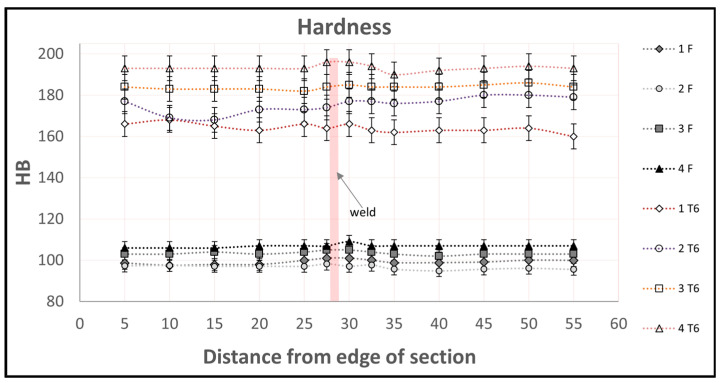
Hardness distributions [HB] on cross-sections of the flat bar 60 × 6 mm: at the top—after heat treatment of the T6 temper; at the bottom—in the F temper.

**Figure 8 materials-16-06429-f008:**
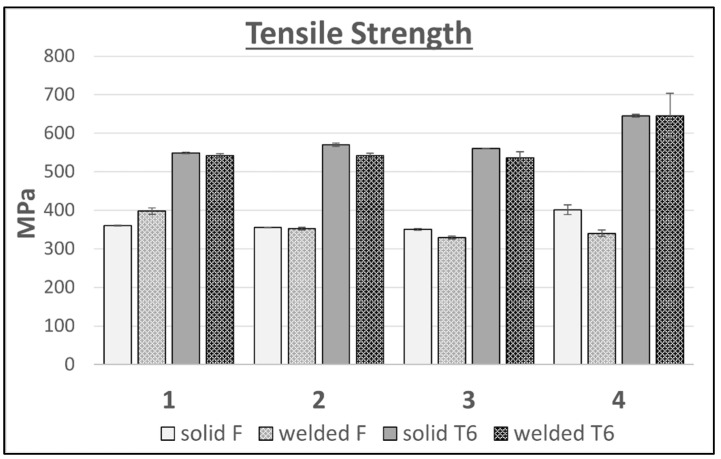
Comparison of the tensile strength results of flat bars with the weld and solid for 4 variants of alloys in the F and T6 tempers.

**Figure 9 materials-16-06429-f009:**
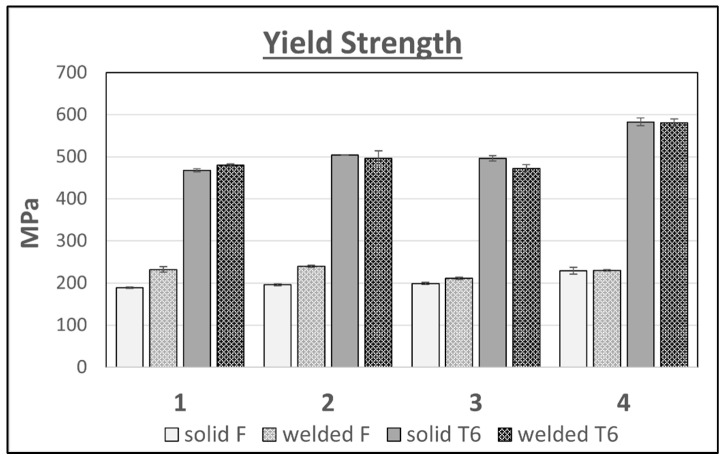
Comparison of the yield strength results of flat bars with the weld and solid for 4 variants of alloys in the F and T6 tempers.

**Figure 10 materials-16-06429-f010:**
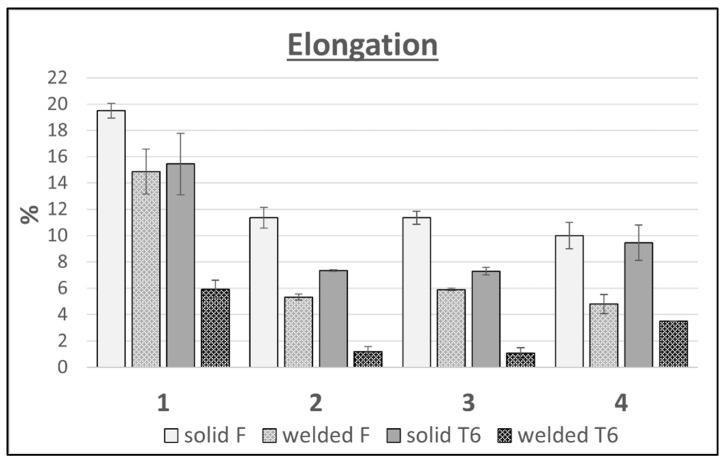
Comparison of the elongations of flat bars with the weld and solid for 4 variants of alloys in the F and T6 tempers.

**Figure 11 materials-16-06429-f011:**
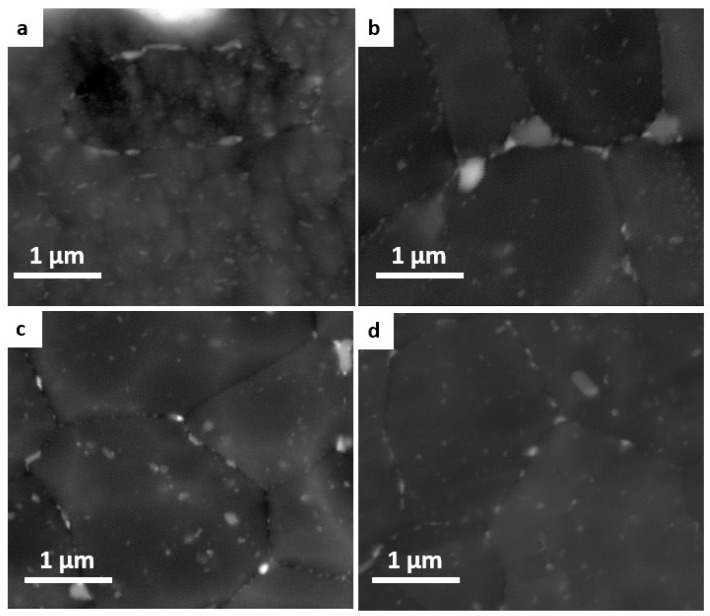
BSE images of investigated alloys: 1 (**a**), 2 (**b**), 3 (**c**), and 4 (**d**).

**Figure 12 materials-16-06429-f012:**
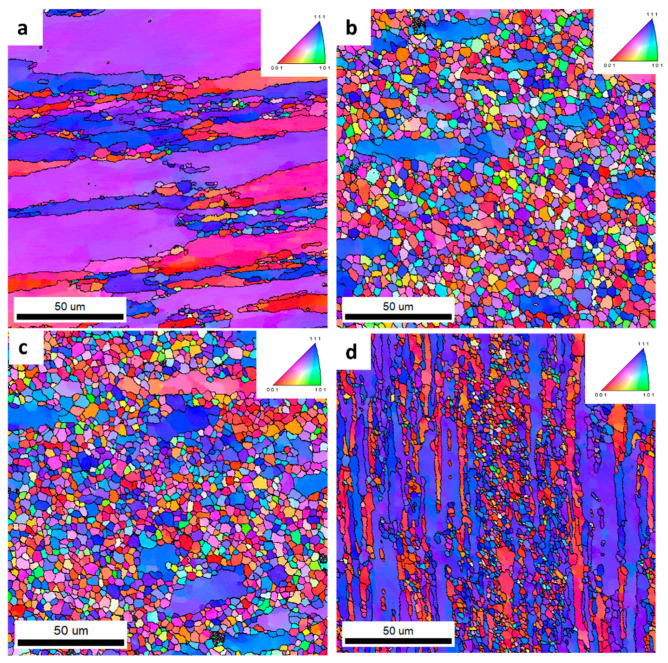
IPF images of investigated alloys: 1 (**a**), 2 (**b**), 3 (**c**), and 4 (**d**)—welded area.

**Figure 13 materials-16-06429-f013:**
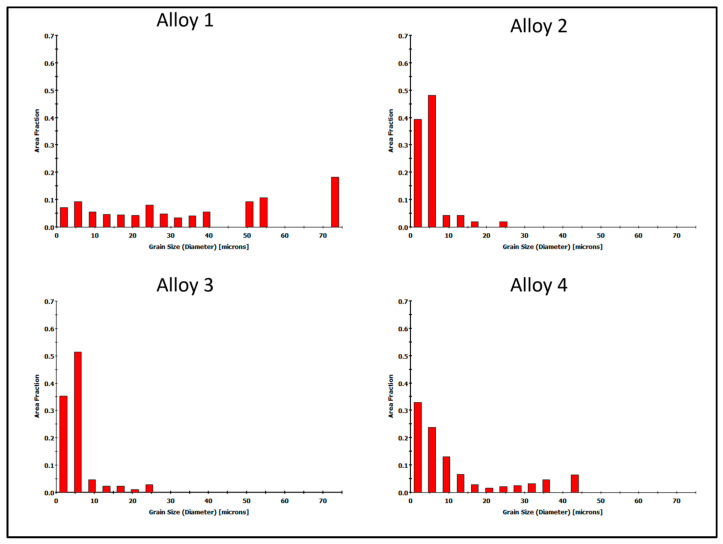
Grain size distributions for investigated alloys—welded area of extruded 60 × 6 mm profile. In red—area fraction of specific grain size.

**Figure 14 materials-16-06429-f014:**
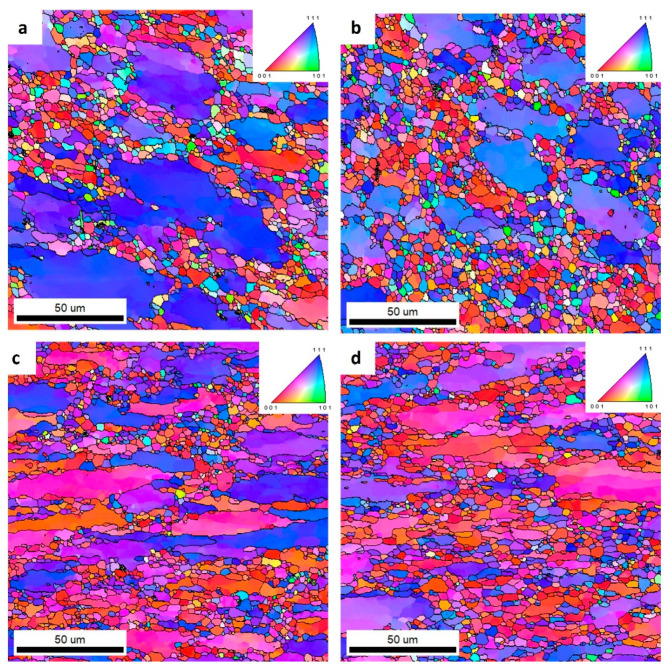
IPF images of investigated alloys: 1 (**a**), 2 (**b**), 3 (**c**), and 4 (**d**)—outside the welded area.

**Figure 15 materials-16-06429-f015:**
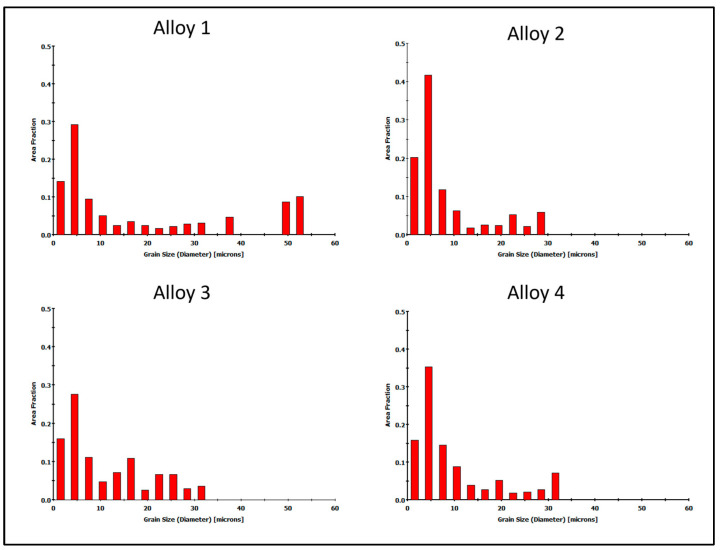
Grain size distributions for investigated alloys—outside the welded area of extruded 60 × 6 mm profile. In red—area fraction of specific grain size.

**Figure 16 materials-16-06429-f016:**
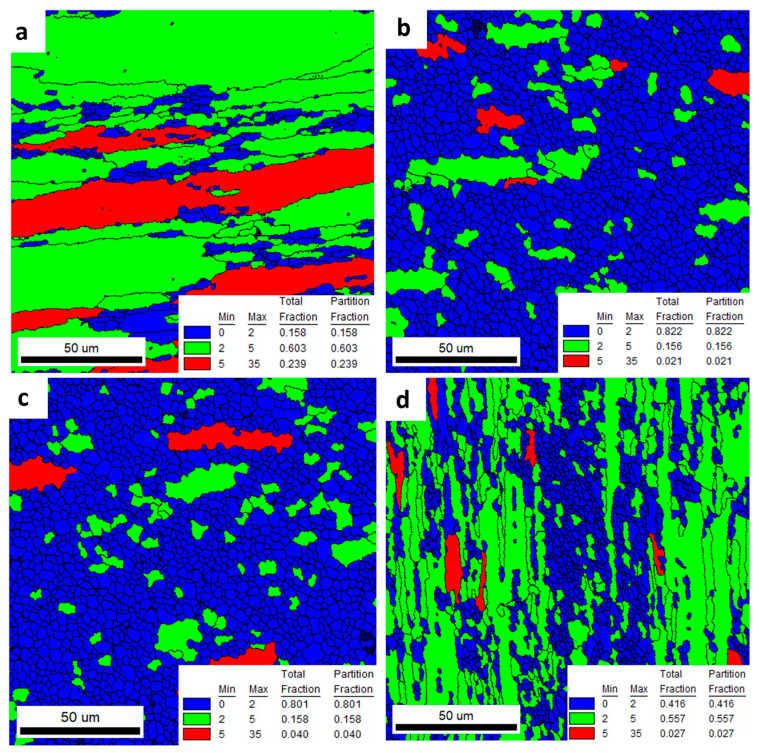
GOS maps of investigated alloys: 1 (**a**), 2 (**b**), 3 (**c**), and 4 (**d**)—welded area.

**Figure 17 materials-16-06429-f017:**
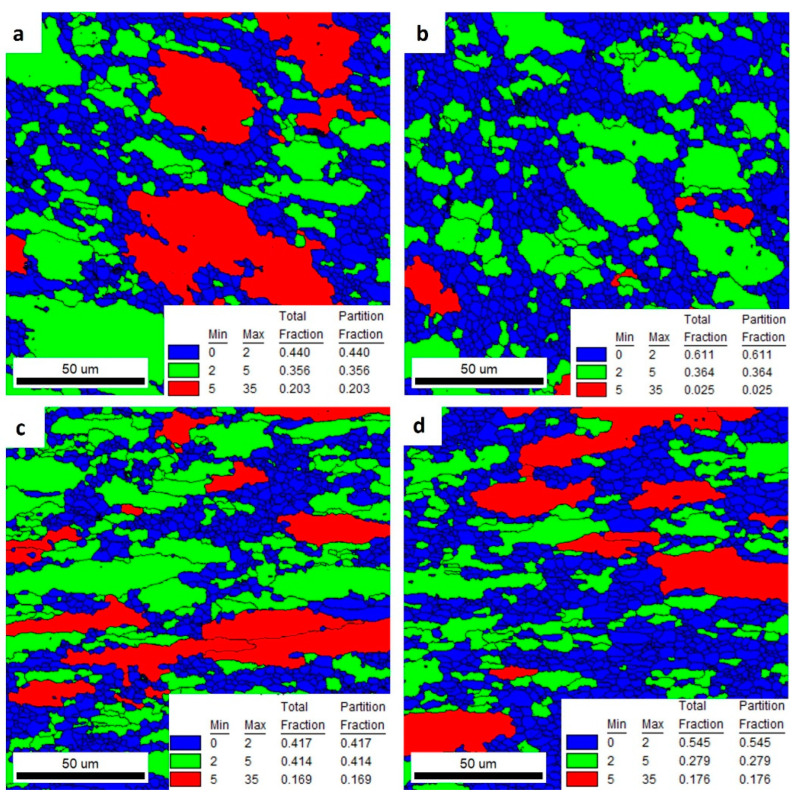
GOS maps of investigated alloys: 1 (**a**), 2 (**b**), 3 (**c**), and 4 (**d**)—outside the welded area.

**Table 1 materials-16-06429-t001:** Chemical compositions of 7075 alloy variants [wt.%].

Alloy No.	Si	Fe	Cu	Mn	Mg	Cr	Zn	Ti	Zr
1	0.08	0.15	1.16	0.00	1.99	0.18	4.97	0.02	0.161
2	0.08	0.15	1.91	0.00	2.33	0.18	5.67	0.02	0.166
3	0.10	0.21	1.53	0.00	2.3	0.18	5.78	0.02	0.151
4	0.10	0.22	1.50	0.00	2.21	0.17	7.76	0.02	0.159

**Table 2 materials-16-06429-t002:** Main alloying elements contents and ratios.

Alloy No.	Cu/Mg	Mg+Zn	Mg+Zn+Cu	Zn/Mg
1	0.58	6.96	8.12	2.50
2	0.82	8	9.91	2.43
3	0.66	8.08	9.61	2.51
4	0.66	9.97	10.47	3.51

**Table 3 materials-16-06429-t003:** Homogenization annealing parameters for tested alloys.

Alloy No.	Heating 1 [°C-h]	Hold Time1 [h]	Heating 2 [°C-min]	Hold Time2 [h]
1	465-10	4	-	-
2	465-10	2	475-15	8
3	465-10	2	475-15	4
4	465-10	12	-	-

**Table 4 materials-16-06429-t004:** Forces registered during the extrusion.

Alloy No.	1	2	3	4
Registered peak extrusion load [MN]	4.11	4.78	4.40	5.06

**Table 5 materials-16-06429-t005:** Heat treatment parameters.

Process	T6 Temper
Solutionizing	temperature 465 °C, hold time 2 h, water quenching
Artificial aging	temperature 120 °C, aging time 24 h

## Data Availability

Not applicable.

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
