# Peer review of "Effects of Zn, Mg, and Cu Content on the Properties and Microstructure of Extrusion-Welded Al–Zn–Mg–Cu Alloys"

_materials, 2023, doi:10.3390/ma16196429_

Round 1

Reviewer 1 Report

1. in the abstract, "with various additions of Mg (22.5%) and Zn (5.5÷8%)" is not consistent with the context; "The highest strength properties after heat treatment were achieved for the alloy with the highest content of Mg and Zn additions, both welded and solid, and are respectively YS=537MPa, TS=567MPa and YS=583MPa, TS=645MPa" is misunderstanding.

2.  The logic of introduction is not well, it is suggested to reorganize the part and adding more review on related research of AlZnMgCu extrusion.

3.  "4. Results and Discussion " should be part 3.

4. "The highest yield resistance, up to 506T, was characteristic of alloy 4, slightly lower, 478T of alloy 3, then 440T of alloy 2 and 411T of alloy 1" is not consistent with Fig.6.

5. "The lowest increase, 48 MPa, was recorded for alloy 4 with a weld, while the highest, 244MPa, was recorded for solid alloy 4" is not consistent with Fig.8 or Fig.9.

6. alloy 1 also showed deformation grains and inhibition of recrystallizaiton, please give explanation.

7. why alloy 2 and 3 had very low elongation with fully recrystallization structure?

8.  "In contrast, alloy 1 had the tensile strength also about 540 MPa, but the elongation was much larger and amount up to 6%. The larger elongation could be induced by the large recrystallized grain size" is not confusive.

9. the format of references are not consistent. 

understandable but need improvement

Author Response

  1. in the abstract, "with various additions of Mg (22.5%) and Zn (5.5÷8%)" is not consistent with the context; "The highest strength properties after heat treatment were achieved for the alloy with the highest content of Mg and Zn additions, both welded and solid, and are respectively YS=537MPa, TS=567MPa and YS=583MPa, TS=645MPa" is misunderstanding.

The absctract has beed edited, errors have been corrected.

  1. The logic of introduction is not well, it is suggested to reorganize the part and adding more review on related research of AlZnMgCu extrusion.

Corrections have been made in the introduction section, descriptions of the process of welding extrusion of AlZnMgCu alloys have been added.

  1. "4. Results and Discussion " should be part 3.

Numeration corrected

  1. "The highest yield resistance, up to 506T, was characteristic of alloy 4, slightly lower, 478T of alloy 3, then 440T of alloy 2 and 411T of alloy 1" is not consistent with Fig.6.

Editorial errors in descriptions have been corrected.

  1. "The lowest increase, 48 MPa, was recorded for alloy 4 with a weld, while the highest, 244MPa, was recorded for solid alloy 4" is not consistent with Fig.8 or Fig.9.

Editorial errors in descriptions have been corrected.

  1. alloy 1 also showed deformation grains and inhibition of recrystallization, please give explanation.

The structure of alloy 1 results from the low content of alloying elements, which translates into low yield resistance. This, in turn, results in a lower level of friction forces in the welding chamber of extrusion die. It resulted in an insufficient temperature increase in the die for dynamic recrystallization to occur. Corrections have been made to the content.

  1. why alloy 2 and 3 had very low elongation with fully recrystallization structure?

According to the authors, in this case, for samples 2 and 3, we observe incomplete recrystallization in the initial phase, and it is not a completely recrystallized structure. The low level of ductility is the result of brittle fracture due to the very fine grain that has developed at the weld border.

The applied heat treatment to the T6 temper was the main reason for reducing the elongation for these variants in the weld zone. Here we have an overlap of 3 mechanisms: the effect of dynamic recovery and recrystallization processes and blocking the migration of grain boundaries during recrystallization by phases containing Zr during the extrusion process altogother with the process of disintegration of the solid solution as a result of T6 heat treatment.

  1. "In contrast, alloy 1 had the tensile strength also about 540 MPa, but the elongation was much larger and amount up to 6%. The larger elongation could be induced by the large recrystallized grain size" is not confusive.

Correction have been made in the text.

  1. the format of references are not consistent.

It has been corrected.

Reviewer 2 Report

1) “However, the literature lacks studies carried out in relation to heat-treated AlZnMgCu alloy extruded with a weld” Why is this study so critically important from the point of view of the authors? What is the peculiarity of the process in terms of phase formation?

2) Why do the authors provide images of furnaces? It will be much more informative to indicate their technical characteristics, as well as the manufacturer and brand. It is also not clear what is shown in figure 1c, add a size mark. In general, it is not informative.

3) “The paper investigate the influence of the different 86 content of main alloy additions on the strength and plasticity of selected AlZ- 87 nMgCu alloys extruded on bridge dies. 88" according to Tables 1 and 2, only the Zn content differs significantly.

4) What is the meaning of image 2? Why does the reader need information a) 3D model, b) completed die insert.

5) “Homogenising parameters were the subject of other research studies 115 conducted [5]. The final annealing conditions are presented in the table below." The formulations presented in study (5) are different from the formulations examined in this article. Please, tell me, is it correct to use the data obtained earlier on alloys of a different composition?

6) Add information about hardness measurement, add more details about it, ball size and load. Why did you choose this method, although the Vickers method is more accurate?

7) “The hardness test was carried out on the 163 cross-section of the flat bars, in the middle of their thickness, with measurement points 164 placed every 5 mm, and in the central zone every 2.5 mm, as shown in Fig. 5." What is the meaning of such accuracy and the most detailed description of the places where hardness measurements were taken in Figure 5? In my opinion, this is redundant, just like Figure 5.

8) “The highest yield resistance, up to 506T, was charac- teristic of alloy 4, slightly lower, 478T of alloy 3, then 440T of alloy 2 and 411T of alloy 1. 185” what does the “T” mean after the cirf?

9) “Figure 6. Press load recorded during 60x6mm section extrusion of 4 aluminum 7XXX series alloys. ."Is it about the alloys you are researching? Or about some other alloys of the 7xxx series?

10) “Figure 7. Hardness distribution [HB] on the cross-section of the flat bar 60x6mm: at 207 the top - after heat treatment to the T6 temper, at the bottom – in the F temper.” confidence interval for given values? Brinell hardness measurement error is significant and your values are very close, which suggests that there is no difference.

11) Figures 11 and 12 are not suitable. It is impossible to identify anything from them.

12) “Numerous studies 256 [21,22] confirm that additives such Zr are inhibitors of recrystallization. The nano-sized dis- 257 persoids formed by Zr can pin down and block the migration of the grain boundaries and, 258 in result, inhibit the recrystallization in AlZnMgCu alloys [22].” why then, according to the presented images, the grain size in alloy 3 (Figure 13c) is much larger than the others? Carefully study the picture and pay attention to the size mark.

13) "All of the alloys reveal intermetallic particles, consisting mainly of such elements as Al, Mg, Cr, Cu and Zn, while the alloys 3 and 4 contain additionally Zr." according to table 1, Zr is contained in all alloys in the same amount

14) Conclusions should be reconsidered.

please review all drawings. they are of very poor quality. present images in high resolution.

Author Response

1) “However, the literature lacks studies carried out in relation to heat-treated AlZnMgCu alloy extruded with a weld” Why is this study so critically important from the point of view of the authors? What is the peculiarity of the process in terms of phase formation?

AlZnMgCu alloys, due to technological difficulties, are rarely used in the extrusion processes on porthole dies. At the same time, the industry is systematically increasing its interest in high-strength Al alloys, also in the form of thin-walled closed profiles. For this reason, as well as due to the lack of relevant publications, there was a need to conduct research on the effect of different levels of the main alloying additives on the extrusion welded AlZnMgCu alloys.

2) Why do the authors provide images of furnaces? It will be much more informative to indicate their technical characteristics, as well as the manufacturer and brand. It is also not clear what is shown in figure 1c, add a size mark. In general, it is not informative.

Figure 1 and the description has been corrected.

3) “The paper investigate the influence of the different 86 content of main alloy additions on the strength and plasticity of selected AlZ- 87 nMgCu alloys extruded on bridge dies. 88" according to Tables 1 and 2, only the Zn content differs significantly.

Aluminum alloys are highly sensitive to the level of the main alloying elements. In case of alloys of the 7xxx series, the difference in Cu or Mg content at the level of 0.5 wt. % may be important from the point of view of the properties of the final material. In addition, the literature indicates that the levels of the alloying elements,as well as the ratios of the individual alloying elements, are important. As previously mentioned, the existing literature presents the results of such tests on extruded materials without a weld. Therefore, it was reasonable to conduct research on the influence of the amount of alloying components on the properties of extrusion welded materials.

4) What is the meaning of image 2? Why does the reader need information a) 3D model, b) completed die insert.

Presentation of the appearance of the die used for research is important from the point of view of describing the methodology, as it is a model matrix only for conducting tests in order to obtain data from the experiment and research material with specific parameters, i.e. a weld along the flat bar. This makes it possible to carry out the weld strength tests planned in the experiment.

Graphics have been improved.

5) “Homogenising parameters were the subject of other research studies 115 conducted [5]. The final annealing conditions are presented in the table below." The formulations presented in study (5) are different from the formulations examined in this article. Please, tell me, is it correct to use the data obtained earlier on alloys of a different composition?

Homogenized alloys in the cited article ([5]- now [13]) were created as part of the same project, they only come from a different casting batch. From the chemical point of view, they are exactly the same materials. In the cited article, there is only a difference in nomenclature, as the 4th variant of the alloy has a chemical composition that falls within the range of alloy 7049 and was named as such.

6) Add information about hardness measurement, add more details about it, ball size and load. Why did you choose this method, although the Vickers method is more accurate?

Ball diameter 2,5mm, main load 31,25kgF.

In our opinion the Brinell test is more suitable for materials with significant differentiation in grain size, which was expected in tested materials. Moreover, we have chosen Brinell method instead of Vickers because of the Brinell lower requirements of surface quality.

7) “The hardness test was carried out on the 163 cross-section of the flat bars, in the middle of their thickness, with measurement points 164 placed every 5 mm, and in the central zone every 2.5 mm, as shown in Fig. 5." What is the meaning of such accuracy and the most detailed description of the places where hardness measurements were taken in Figure 5? In my opinion, this is redundant, just like Figure 5.

The research was conducted on extrusion welded materials. It was important to verify whether there was a difference in properties across the examined materials. This was done by the hardness test.

Increasing the amount of measurement points in the central zone was intended to increase precision in determining the zone of influence of the weld on properties.

8) “The highest yield resistance, up to 506T, was charac- teristic of alloy 4, slightly lower, 478T of alloy 3, then 440T of alloy 2 and 411T of alloy 1. 185” what does the “T” mean after the cirf?

Units changed to MN, descriptions have been corrected.

9) “Figure 6. Press load recorded during 60x6mm section extrusion of 4 aluminum 7XXX series alloys. ."Is it about the alloys you are researching? Or about some other alloys of the 7xxx series?

Descriptions have been improved.

10) “Figure 7. Hardness distribution [HB] on the cross-section of the flat bar 60x6mm: at 207 the top - after heat treatment to the T6 temper, at the bottom – in the F temper.” confidence interval for given values? Brinell hardness measurement error is significant, and your values are very close, which suggests that there is no difference.

There was a minor difference between the investigated alloys in F temper, however after the heat treatment to T6 temper the differences are noticeable. The Figure 7 also shows the differentiation between the investigated alloys in F and T6 temper. Confidence interval was added.

11) Figures 11 and 12 are not suitable. It is impossible to identify anything from them.

Figure 11 have been changed, Figure 12 has been deleted as irrelevant.

12) “Numerous studies 256 [21,22] confirm that additives such Zr are inhibitors of recrystallization. The nano-sized dis- 257 persoids formed by Zr can pin down and block the migration of the grain boundaries and, 258 in result, inhibit the recrystallization in AlZnMgCu alloys [22].” why then, according to the presented images, the grain size in alloy 3 (Figure 13c) is much larger than the others? Carefully study the picture and pay attention to the size mark.

The EBSD image of alloy 3 was rescaled. As a result, it was shown that the size of the recrystallized grains is similar to that of alloy 2. Corrections have been made in the content.

13) "All of the alloys reveal intermetallic particles, consisting mainly of such elements as Al, Mg, Cr, Cu and Zn, while the alloys 3 and 4 contain additionally Zr." according to table 1, Zr is contained in all alloys in the same amount

Corrections in the content have been made.

14) Conclusions should be reconsidered.

Conclusions have been corrected

please review all drawings. they are of very poor quality. present images in high resolution.

All graphics have been changed to higher resolution ones. High resolution graphics have been attached in the supplementary file.

Reviewer 3 Report

The authors investigated the effect of Zn, Mg, and Cu content on the properties and microstructure of extrusion-welded AlZnMgCu alloys. I have some comments and suggestions to improve the quality of current manuscript as below:

1. English should be improved. Please avoid using the abbreviation in abstract. The abbreviation should be defined before using.

2. The abstract should be rewritten to address what authors done and obtained results. 

3. The authors mentioned "The highest strength properties after heat treatment were achieved for the alloy with the highest content of Mg and Zn additions", please provide exactly values of optimized Mg and Zn content.

4. The authors mentioned to the formation of precipitates such as MgZn2, Al3Zr, etc.. but there is no supported results. Therefore, HRTEM or XRD should be conducted and presented.

5.  Grain size distributions of sample should be made and presented.

English should be improved.

Author Response

  1. English should be improved. Please avoid using the abbreviation in abstract. The abbreviation should be defined before using.

The article have been corrected by the english-fluent editor. The absctract has beed edited, errors have been corrected.

 The abstract should be rewritten to address what authors done and obtained results.

The absctract has beed edited, errors have been corrected.

  1. The authors mentioned "The highest strength properties after heat treatment were achieved for the alloy with the highest content of Mg and Zn additions", please provide exactly values of optimized Mg and Zn content.

The absctract has beed edited, errors have been corrected.

  1. The authors mentioned to the formation of precipitates such as MgZn2, Al3Zr, etc.. but there is no supported results. Therefore, HRTEM or XRD should be conducted and presented.

The authors base their descriptions on the existing knowledge and on the literature data.

Due to the failure of the cooling system, it was not possible to supplement the research with XRD analysis.

  1. Grain size distributions of sample should be made and presented.

Grain size distribution histograms have been added, descriptions have been corrected.

Round 2

Reviewer 1 Report

Significant modification is made. Only one suggestion for further modification.

1. "A clear decrease in elongation in the weld areas is noticeable for each of the tested alloy variants", the explanation should be given according to microstructures.

Author Response

The description related to the dependence of elongation on microstructure for the case of alloys 2, 3 and 4 has been supplemented, starting from line 359.
The authors believe that the decrease in plasticity in the weld is the result of both the fine-grained microstructure and the precipitation strengthening, the greater the higher the content of alloying elements.

Reviewer 3 Report

The manuscript could be accepted in its current form.

 Minor editing of English language required

Author Response

Due to the short response time, the authors did not manage to obtain language corrections from a english native speeking researcher.